# Topological layer Hall effect in two-dimensional type-I multiferroic heterostructure

Wenhui Du, Kaiying Dou, Xinru Li, Ying Dai ✉, Zeyan Wang ✉, Baibiao Huang & Yandong Ma ✉

Magnetic skyrmion and layer physics have attracted considerable interest for their significance in fundamental research and practical device applications. Here, through symmetry and model analysis, we propose a mechanism for coupling magnetic skyrmion and layer physics in two-dimensional type-I multiferroic heterostructure, which generates the concept of topological layer Hall effect. Distinct from the existing layer Hall effects that are all driven by momentum-space Berry phase relied on fine-tuned bands, topological layer Hall effect correlates to the layer-polarized real-space Berry physics from noncoplanar spin textures of layer-locked magnetic skyrmion with nontrivial topology. Such layer-polarized real-space Berry physics acts as equivalent electromagnetic field and forces conduction electrons to transversely deflect to specific boundary of one given layer, yielding the anomalous Hall conductivity and thus topological layer Hall effect. Moreover, magnetoelectric coupling can enforce topological layer Hall effect being effectively controllable through ferroelectricity and magnetism. Using first-principles calculations and atomic spin model simulations, we also demonstrate this mechanism in two-dimensional multiferroic heterostructure of $CrInSe_3$/$In_2S_3$/$CrInSe_3$. Our study greatly enriches the researches on magnetic skyrmion and layer Hall effect.

Magnetic skyrmions are swirling spin textures with non-trivial topology characterized by a quantized topological number in real space[1–3]. Since the first experimental observations in bulk non-centrosymmetric magnets[4,5] and thin films[6], these spin-chiral objects have been a prominent topic of condensed matter physics owing to a variety of compelling characteristics, e.g., nanoscale dimension, stability with topological protection and solitonic nature[7–19]. These exotic features not only hold significant promise for exploring nontrivial topological physics but also offer unparalleled prospects for future spintronics applications. In view of advancing more fundamentals and translating these exotic features into skyrmion-based spintronics devices with superior performance, it is crucial to achieve effective control of skyrmion physics, such as coupling skyrmion properties to charge and spin degrees of freedom[20–27].

Recently, layer degree has been identified as an extra degree of freedom of electrons in addition to charge and spin[28–30]. The emergence of layer degree reveals a remarkable type of Hall effect, i.e., layer Hall effect (LHE), which characterizes the layer-polarized transverse deflections of electrons[31]. Leveraging this unique attribute, LHE exhibits properties that partially resemble conventional Hall effects, while also presenting distinct merits that have received significant attention in the field of spintronics, especially for information processing and storage[32–34]. LHE is first experimentally observed in antiferromagnetic (AFM) axion insulator $MnBi_2Te_4$ thin films[31], and subsequently

School of Physics, State Key Laboratory of Crystal Materials, Shandong University, Jinan, China. ✉e-mail: daiy60@sina.com; wangzeyan@sdu.edu.cn; yandong.ma@sdu.edu.cn

reported in several asymmetric valleytronic van der Waals bilayers[35–37]. Yet, to the best of our knowledge, all these few existing works on LHE are rooted in the paradigm of momentum-space Berry phase[31–39], which necessarily requires fine-tuned bands that are not easily satisfied. This raises an outstanding challenge for the development of LHE. To conquer this issue, one must go beyond the existing paradigm.

In this work, we show that magnetic skyrmions can be coupled to layer physics in a two-dimensional (2D) type-I multiferroic heterostructure, giving rise to a novel phenomenon of topological layer Hall effect (TLHE). Via symmetry and model analysis, we reveal that TLHE originates from the layer-polarized real-space Berry physics, which arises from a layer-locked magnetic skyrmion with nontrivial topology. This mechanism is fundamentally distinct from the existing LHEs that are all based on the paradigm of momentum-space Berry phase dependent on fine-tuned bands. Because of the layer-polarized real-space Berry physics acting as an equivalent electromagnetic field, the conduction electrons transversely deflect to a specific boundary of one given layer, which results in the anomalous Hall conductivity as well as TLHE. Besides, due to the particular magnetoelectric coupling, TLHE can be effectively controlled by ferroelectricity and magnetism. By means of first-principles calculations and atomic spin model simulations, this mechanism is further demonstrated in a 2D multiferroic heterostructure of $CrInSe_3/In_2S_3/CrInSe_3$. Our findings not only open a new direction for research in the manipulation of magnetic skyrmions but also overcome the restrictions of LHE.

## Results

### Coupling magnetic skyrmion and layer physics

Since our target is to generate the coupling between magnetic skyrmion and layer physics, the systems studied here should be multilayers with layer polarization and time-reversal symmetry breaking. We also require the layer polarized physics to be reversible. These requirements naturally point to a 2D multiferroic heterostructure. Inspired by these insights, we here focus on a 2D type-I multiferroic heterostructure with sandwiching one ferroelectric (FE) single-layer between two ferromagnetic (FM) single-layers (referred to as A-layer and B-layer) with large Dzyaloshinskii–Moriya interaction (DMI). As illustrated in the left panel of Fig. 1a, due to the existence of out-of-plane polarization ($P_+$ phase), A- and B-layers experience distinct proximity effects[40] from the FE layer. Quite naturally, this would result in differences in magnetic properties of the two FM layers, including magnetic moments ($M \neq M'$) and magnetic interactions ($J \neq J', K \neq K', D \neq D'$). Upon reversing the electric polarization, the

properties of these two FM layers would be exchanged. Furthermore, due to the relatively large spatial separation between these two FM layers, the interlayer exchange interaction would be significantly weaker as compared with the intralayer interaction. In this regard, different topological spin textures [e.g., skyrmion (SkX) and bimeron] might be expected in these two FM layers. Without loss of generality, as illustrated in the top-left panel of Fig. 1c, we assume that the A-layer favors SkX phase, while the B-layer presents a trivial FM phase [referred to as ($P_+$, $M_+$) phase].

Physically, when a conduction electron couples with non-coplanar spin textures in the A-layer, its trajectory is affected by the background spin texture, endowing it with quantum-mechanical phase, i.e., real-space Berry curvature. Similar to the momentum-space Berry curvature[41], real-space Berry curvature can affect the charge transport in a similar way as external magnetic flux does. Specifically, real-space Berry curvature acts as an effective magnetic field on the conduction electrons, wherein the equivalent magnetic field $\mathbf{B}^e$ can be expressed as[42]

$$\mathbf{B}^e = \frac{\hbar}{2e}\hat{n} \cdot (\partial_x \hat{n} \times \partial_y \hat{n})\mathbf{z} \qquad (1)$$

Here, $\hat{n}(\mathbf{r}, t) = \mathbf{m}/|\mathbf{m}|$ describes the spin texture, $\partial_x = \partial/\partial x$ and $\partial_y = \partial/\partial y$. This gives rise to an effective Lorentz force of $\mathbf{F} = -e(\dot{\mathbf{r}} \times \mathbf{B}^e)$ exerting on the conduction electrons, leading to their transverse deflections, as shown in Fig. 1b. While for B-layer, the absence of the real-space Berry curvature ensures that the conduction electrons will move in a straight line. Therefore, for ($P_+$, $M_+$) phase, due to the layer-polarized real-space Berry curvature, spin-up conduction electrons would accumulate at right edge of A-layer, giving rise to the concept of TLHE; see top-left panel of Fig. 1c. We wish to stress that the TLHE is based on the paradigm of the layer-polarized real-space Berry physics raised from layer-locked noncoplanar spin textures of magnetic skyrmion with nontrivial topology, which is in sharp contrast to the existing LHEs that all result from the paradigm of momentum-space Berry phase dependent on fine-tuned bands[31–39]. Meanwhile, TLHE introduces a layer degree of freedom as a tunable parameter for transport, which distinguishes it from conventional Hall effects (e.g., quantum anomalous Hall effect[43] and anomalous valley Hall effect[44]). And as compared with LHE, TLHE represents an important step towards spatial engineering of topological spintronics.

Upon reversing the FE polarization via FE transition, ($P_+$, $M_+$) phase is transformed into ($P_-$, $M_+$) phase. Accompanied by this

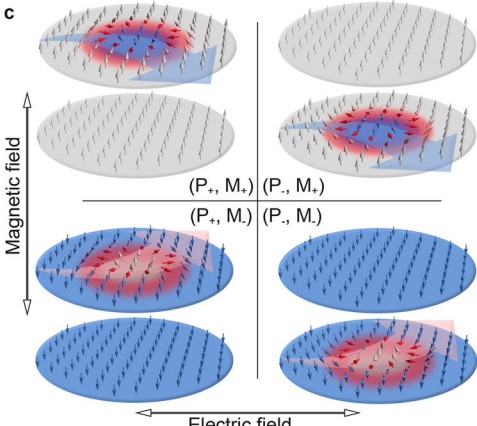

**Fig. 1 | Coupling magnetic skyrmion and layer physics in 2D multiferroic heterostructure. a** Schematic of the coupling between magnetic skyrmion and layer physics in a 2D type-I multiferroic heterostructure. In **a**, $M(M'), J(J'), K(K')$, and $D(D')$ represent magnetic moment, intralayer exchange interaction, single-ion anisotropy and intralayer DMI of A(B)-layer, respectively, for $P_+$ phase. **b** Schematic of interaction between conduction electrons and magnetic skyrmion. **c** Schematic of TLHE in 2D multiferroic heterostructure with different combinations of electric polarization (P) and magnetization orientation (M).

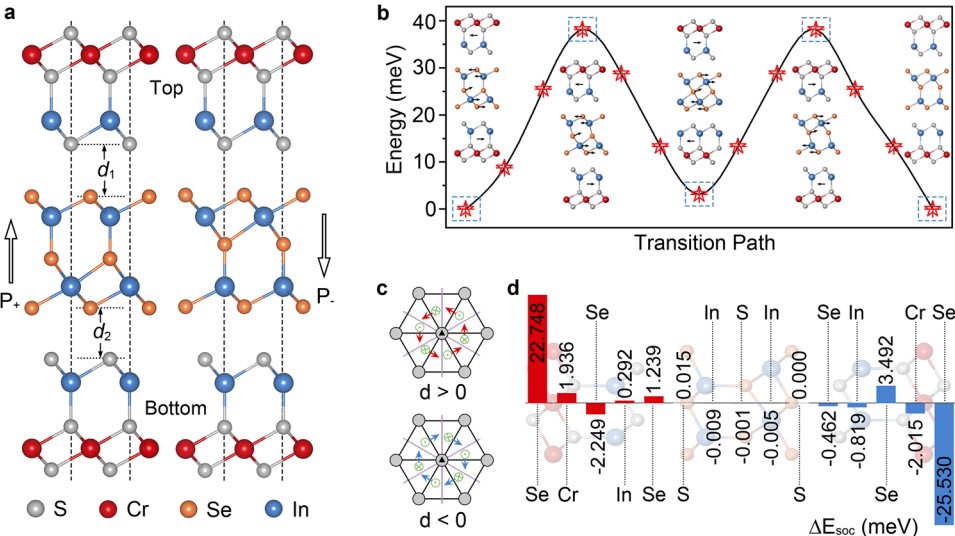

**Fig. 2 | Structures, FE properties, and DMI of CrInSe₃/In₂S₃/CrInSe₃. a** Crystal structures of CrInSe₃/In₂S₃/CrInSe₃ heterostructure under P₊ and P₋ phases. **b** FE transition pathway and energy barrier between P₊ and P₋ phases. **c** Schematic diagrams of the DMI vectors between the nearest-neighboring Cr atoms. In (**c**), gray balls, black triangles, and purple lines represent nearest-neighboring Cr atoms, threefold out-of-plane rotation axes, and mirror planes, respectively. In-plane and out-of-plane DMI vectors between the nearest-neighboring Cr atoms are indicated by the red/blue arrows and dot/cross symbols, respectively. **d** Atomic-layer-projected SOC energy ($\Delta E_{\text{SOC}}$) for P₊ phase.

process, as shown in the right panel of Fig. 1a, the situations for A- and B-layers are exchanged, i.e., A-layer displays a trivial FM phase, while B-layer exhibits SkX phase; see top-right panel of Fig. 1c. This suggests that the layer-polarized real-space Berry curvature transforms to B-layer. When conduction electrons pass through the system, the spin-up electrons transversely deflect to the right edge of the B-layer. It is important to note that due to the antiparallel stacking pattern, the DMI chirality in the B-layer is opposite to that in the A-layer. However, for magnetic skyrmions, the DMI chirality reversal does not alter the sign of the Lorentz force[45]. In addition, since the real-space Berry curvature induced Lorentz force is determined by $N_{x,y}(\mathbf{r}) = \hat{\mathbf{n}} \cdot \left( \frac{\partial \hat{\mathbf{n}}}{\partial x} \times \frac{\partial \hat{\mathbf{n}}}{\partial y} \right)$, when reversing spin orientation [i.e., (P₊, M₋) and (P₋, M₋) phases], the Lorentz forces acting on conduction electrons are also reversed. This leads to the reversal of deflection directions of conduction electrons, as shown in the bottom panels of Fig. 1c. Accordingly, due to the particular magnetoelectric coupling, TLHE can be effectively controlled by ferroelectricity and magnetism.

## Magnetic interactions in CrInSe₃/In₂S₃/CrInSe₃

In the following, using first-principles calculations and atomic spin model simulations, we demonstrate the idea of TLHE using a multiferroic heterostructure of CrInSe₃/In₂S₃/CrInSe₃ as an example. It consists of two antiparallelly stacked CrInSe₃ single layers sandwiching one α-In₂S₃ single layer. While single-layer CrInSe₃ harbors skyrmion physics[46], single-layer α-In₂S₃ is known for its out-of-plane FE polarization[47]. The lattice constants of CrInSe₃ and α-In₂S₃ are optimized to be 3.92 and 3.94 Å, respectively, yielding a lattice mismatch of only 0.7%. Given this small lattice mismatch and the van der Waals layered nature of both CrInSe₃ and In₂S₃, the fabrication of CrInSe₃/In₂S₃/CrInSe₃ heterostructure is practically feasible. Recent advances in mechanical exfoliation, van der Waals stacking, and epitaxial growth techniques such as chemical vapor deposition and molecular beam epitaxy provide promising pathways for realizing such layered structures experimentally[48]. Considering the switchable polarization of α-In₂S₃, the heterostructure naturally displays two types: the first one with polarization pointing to top CrInSe₃ (P₊), and the second one with polarization pointing to bottom CrInSe₃ (P₋). For each type, nine high-symmetry stacking patterns are considered (Supplementary Fig. 1), and the energy minimum configurations are shown in Fig. 2a.

Because of the presence of electric polarization, the two van der Waals interfaces of CrInSe₃/In₂S₃/CrInSe₃ are different from each other. In detail, for P₊ (P₋) phase, the upper and lower interlayer distances are optimized to be $d_1 = 3.05$ (2.95) Å and $d_2 = 2.95$ (3.05) Å, respectively. Furthermore, as shown in Supplementary Fig. 2, the differential charge densities at the two interfaces are markedly different. This suggests the distinct proximity effects experienced by the top and bottom layers of CrInSe₃. The electric polarizations for P₊ and P₋ phases are calculated to be 3.186 and −3.186 pC/m, respectively, showing equal magnitudes, but opposite signs. Therefore, P₊ and P₋ phases can be regarded as two energetically degenerate FE states of CrInSe₃/In₂S₃/CrInSe₃. To estimate the feasibility of ferroelectricity in CrInSe₃/In₂S₃/CrInSe₃, we investigate the FE switching process using the climbing image nudged elastic band (CI-NEB) method. As shown in Fig. 2b, the energy barrier for FE transition is calculated to be 38 meV per formula unit. For comparison, the FE transition barrier of monolayer In₂Se₃ has been theoretically estimated to be 66 meV[49], and the corresponding experimental electric field required for polarization switching is about 200 kV/cm[50]. Based on this fact, the electric field needed to reverse the polarization of In₂S₃ in the CrInSe₃/In₂S₃/CrInSe₃ heterostructure is expected to be below 200 kV/cm, confirming the feasibility of realizing ferroelectricity in this system.

In addition to ferroelectricity, CrInSe₃/In₂S₃/CrInSe₃ also exhibits spin polarization. In P₊ phase, the magnetic moments are primarily localized on Cr_top and Cr_bot atoms, which are found to be 3.473 and 3.467 μ_B, respectively. Upon transforming to P₋ phase, the magnetic moments of Cr_top and Cr_bot atoms are reversed, suggesting an intriguing magnetoelectric coupling effect. While magnetic proximity effects could theoretically induce weak magnetic moments in the In₂S₃ layer, our calculations show these moments are negligible (<0.001 μ_B/atom) compared to those in the Cr atom. This three-orders-of-magnitude difference confirms that In₂S₃ cannot support topological spin textures. Therefore, our analysis focuses exclusively on skyrmion formation in the CrInSe₃ layers.

**Table 1 | Magnetic parameters (in meV) for $P_+$ and $P_-$ phases of CrInSe$_3$/In$_2$S$_3$/CrInSe$_3$**

| | $J_1^{top}$ | $J_2^{top}$ | $J_1^{bot}$ | $J_2^{bot}$ | $J_1^{inter}$ | $J_2^{inter}$ | $d_\parallel^{top}$ | $d_\parallel^{bot}$ | $K^{top}$ | $K^{bot}$ |
|---|---|---|---|---|---|---|---|---|---|---|
| $P_+$ | 28.672 | −2.094 | 29.729 | −1.866 | 0.003 | 0.004 | 2.006 | −2.048 | 0.422 | 0.673 |
| $P_-$ | 29.729 | −1.866 | 28.672 | −2.094 | 0.003 | 0.004 | 2.048 | −2.006 | 0.673 | 0.422 |

For further exploring the magnetic properties of CrInSe$_3$/In$_2$S$_3$/CrInSe$_3$, we introduce the following Heisenberg spin Hamiltonian:

$$H = H_{intra}^{top} + H_{intra}^{bot} + H_{inter} \qquad (2)$$

Here, $H_{intra}^{top(bot)}$ represents the intralayer magnetic interaction in top (bottom) CrInSe$_3$ layer, while $H_{inter}$ denotes the interlayer magnetic interaction. Specifically,

$$\begin{aligned}
H_{intra}^{top(bot)} = &-J_1 \sum_{\langle i,j \rangle} \mathbf{S}_i \cdot \mathbf{S}_j - J_2 \sum_{\langle i',j' \rangle} \mathbf{S}_{i'} \cdot \mathbf{S}_{j'} - K \sum_i (\mathbf{S}_i^z)^2 \\
&- \sum_{\langle i,j \rangle} \mathbf{D}_{ij} \cdot (\mathbf{S}_i \times \mathbf{S}_j) - mB_z \sum_i \mathbf{S}_i^z
\end{aligned} \qquad (3)$$

$$H_{inter} = -J_1^{inter} \sum_{\langle i \in top, j \in bot \rangle} \mathbf{S}_i \cdot \mathbf{S}_j - J_2^{inter} \sum_{\langle i' \in top, j' \in bot \rangle} \mathbf{S}_{i'} \cdot \mathbf{S}_{j'} \qquad (4)$$

Here $\mathbf{S}_i$ is the unit vector of the local spin at the $i^{th}$ Cr site. $\langle i,j \rangle$ and $\langle i',j' \rangle$ donate nearest-neighboring (NN) and next-NN (NNN) sites, respectively. The parameters $J_1$ ($J_1^{inter}$) and $J_2$ ($J_2^{inter}$) correspond to the NN and NNN intralayer (interlayer) exchange interactions ($J > 0$ indicates FM coupling), respectively. $K$ and $\mathbf{D}_{ij}$ represent the magnetic parameters for single-ion anisotropy and intralayer DMI, respectively. The last term of $H_{intra}^{top(bot)}$ is Zeeman effect, where $m$ and $B_z$ represent the magnetic moment and external magnetic field along $z$ direction, respectively.

The obtained magnetic parameters from Eq. (2) are summarized in Table 1. For top CrInSe$_3$ layer in $P_+$ phase, the intralayer exchange interactions are calculated to be $J_1^{top} = 28.672$ meV and $J_2^{top} = -2.094$ meV, respectively, suggesting that NN interaction favors FM coupling, while NNN interaction prefers AFM coupling. Considering the large difference between $J_2^{top}$ and $J_1^{top}$, FM coupling dominates the intralayer magnetic exchange. For the bottom CrInSe$_3$ layer in $P_+$ phase, $J_1^{bot}$ and $J_2^{bot}$ follow a similar trend, but exhibit slightly different coupling strengths from the top CrInSe$_3$ layer. For single-ion anisotropy, it is calculated to be 0.422 and 0.673 meV for the two layers, respectively, which corresponds to the preference of out-of-plane magnetization orientation. Obviously, the difference in single-ion anisotropy is significant, which is in contrast to that of the intralayer exchange interaction. For interlayer exchange interactions, $J_1^{inter}$ and $J_2^{inter}$ exhibit positive values, indicating that FM coupling is preferred. Additionally, it can be seen that $J_1^{inter}$ and $J_2^{inter}$ are four orders of magnitude smaller than $J_1$, suggesting that the exchange interaction between two CrInSe$_3$ layers is remarkably weak with respect to that within each layer. Supplementary Fig. 3 presents the band structure of $P_+$ phase considering spin–orbit coupling (SOC). It can be seen that both the spin-up and spin-down bands cross the Fermi level, confirming the 2D FM metallic nature.

According to the Moriya's rule[51], the DMI vector $\mathbf{D}_{ij}$ for NN Cr atoms is perpendicular to their bond owning to the $C_{3v}$ symmetry, as shown in Fig. 2c. It can be expressed as $\mathbf{D}_{ij} = d_\parallel (\mathbf{z} \times \mathbf{u}_{ij}) + d_\perp \mathbf{z}$. Here, $\mathbf{z}$ is the unit vector along $z$-direction and $\mathbf{u}_{ij}$ is the unit vector pointing from sites $i$ to $j$. Since the total contribution of out-of-plane component $d_\perp$ is negligible, we focus on in-plane component $d_\parallel$. To calculate in-plane component $d_\parallel$, we constructed a $1 \times 4$ supercell and set four different spin-spiral configurations, as illustrated in Supplementary Fig. 4. For $P_+$ phase, in-plane components $d_\parallel$ are calculated to be 2.006 meV for the

top layer and −2.048 meV for the bottom layer. The opposite signs of DMI indicate that the DMI chirality in these two layers is opposite. To further investigate the origin of the large DMI in $P_+$ phase, we examined the atomic-layer-resolved SOC energy difference $\Delta E_{soc}$, defined as the energy difference between right- and left-hand spin-spiral configurations, as shown in Supplementary Fig. 5. As shown in Fig. 2d, the DMI is primarily contributed by the neighboring Se atom, while the magnetic Cr atom makes little contribution, suggesting a typical Fert–Levy mechanism[52,53].

When transforming $P_+$ into $P_-$ phases under ferroelectricity, the middle In atomic layer of $\alpha$-In$_2$S$_3$ shifts towards the top CrInSe$_3$ layer, which exchanges the local environments of Cr$_{top}$ and Cr$_{bot}$ atoms. Consequently, the magnetic exchange interactions and single-ion anisotropy in the top CrInSe$_3$ layer are swapped with those in the bottom CrInSe$_3$ layer. Yet, as listed in Table 1, the exchange interactions between two layers remain unchanged. Meanwhile, the FE transformation alters the DMI strength of two layers while preserving their chirality. It is important to note that, in each FE phase, the strengths of magnetic interactions in the top and bottom layers are different, particularly regarding the single-ion anisotropy.

The asymmetric nature of DMI and single-ion anisotropy can be attributed to the different proximity effects exerted by $\alpha$-In$_2$S$_3$. As illustrated in Supplementary Fig. 2, the charge transfers from the bottom CrInSe$_3$ layer to the In$_2$S$_3$ layer, resulting in charge redistributions around Cr atoms and SOC-active Se atoms in the bottom CrInSe$_3$ layer. In contrast, no such charge transfer is observed between the top CrInSe$_3$ layer and In$_2$S$_3$ layer, indicating a much weaker proximity effect. This leads to distinct electronic environments near the Fermi level for the top and bottom CrInSe$_3$ layers, particularly in terms of orbital occupation (see Supplementary Fig. 6). It should be noted that both DMI and single-ion anisotropy originate from SOC-mediated interactions between occupied and unoccupied states near the Fermi level[54,55]. Consequently, the SOC-mediated interaction between occupied and unoccupied states becomes layer-dependent, giving rise to the observed asymmetry in both DMI and single-ion anisotropy between the two CrInSe$_3$ layers.

## Topological layer Hall effect

Due to the differences in magnetic properties, we can expect the emergence of distinct spin textures in the top and bottom layers of CrInSe$_3$/In$_2$S$_3$/CrInSe$_3$, potentially enabling the realization of the TLHE. To verify this scenario, we perform atomic spin model simulations to explore the possible spin textures in CrInSe$_3$/In$_2$S$_3$/CrInSe$_3$ based on the first-principles parameterized Hamiltonian and Landau–Lifshitz–Gilbert (LLG) equation. For the ($P_+$, $M_+$) phase, as shown in Fig. 3a, isolated magnetic skyrmions emerge in both top and bottom layers, indicating that both layers exhibit the SkX phase in the absence of an external magnetic field. However, the magnetic skyrmions in the top and bottom layers show opposite helicities. This helicity difference stems from the opposite DMI chirality between the two layers. Meanwhile, as shown in Fig. 3a and Supplementary Fig. 7, the skyrmion sizes in the two layers are significantly distinct, which is attributed to the differences in the strength of magnetic interactions.

Figure 3b illustrates the evolution of topological charge $Q$ in the top and bottom layers of ($P_+$, $M_+$) as a function of magnetic field. It is evident that the density of skyrmions in the top and bottom layers exhibits different responses to the magnetic field. Specifically, in the

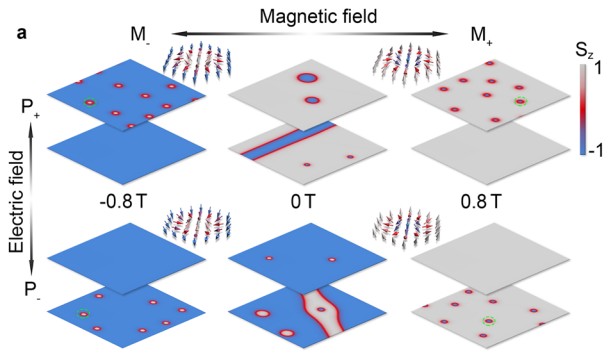

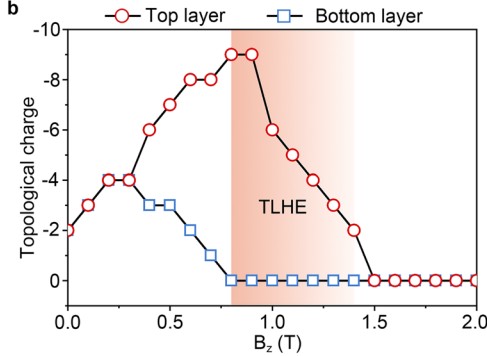

**Fig. 3 | Spin textures of CrInSe$_3$/In$_2$S$_3$/CrInSe$_3$. a** Spin textures of CrInSe$_3$/In$_2$S$_3$/CrInSe$_3$ with different combinations of electric polarization (P) and magnetization orientation (M). Colors in (**a**) indicate the out-of-plane magnetization components. **b** Evolution of topological charge $Q$ of the two layers in P$_+$ phase of CrInSe$_3$/In$_2$S$_3$/ CrInSe$_3$ as functions of external magnetic field. The orange region in (**b**) indicates the magnetic field range enabling the realization of the topological layer Hall effect (TLHE).

top layer, as the magnetic field increases from 0 to 0.8 T, it gradually increases and reaches a maximum of $1.5 \times 10^{-3}$ per nm$^2$ (9 per supercell) at 0.8 T. While in the bottom layer, it first increases under a magnetic field of 0–0.3 T, but with further increasing magnetic field, it decreases and eventually drops to zero at 0.8 T. Therefore, under a magnetic field of 0.8 T, the bottom layer reverses to a trivial FM phase, while the top layer remains in SkX phase, as shown in Fig. 3a. This results in the layer-contrasted topological magnetism, i.e., the layer degree of freedom is coupled with magnetic skyrmion. Upon reversing the electric polarization through FE switching to form (P$_-$, M$_+$), the skyrmion properties, including size and density in the top layer, are swapped with those in the bottom layer. As shown in Fig. 3a, under a magnetic field of 0.8 T, the spin texture in the top layer exhibits FM phase, while the bottom layer shifts to SkX phase. Thus, the layer-contrasted topological magnetism can be switched through ferroelectricity. On the other hand, by reversing the magnetic field to form (P$_+$, M$_-$) and (P$_-$, M$_-$), as shown in Fig. 3a, the core spin orientation of magnetic skyrmions is also reversed, resulting in the change of topological charge from $Q = -1$ to 1. In the following, if otherwise specified, we focus on these cases with layer-contrasting skyrmion physics.

Due to the layer-contrasting skyrmion physics, when conduction electrons flow through CrInSe$_3$/In$_2$S$_3$/CrInSe$_3$, they would experience different real-space Berry curvature in the top and bottom layers. It suggests the formation of layer-contrasting real-space Berry physics in CrInSe$_3$/In$_2$S$_3$/CrInSe$_3$, which can guarantee the intriguing TLHE. For example, in (P$_+$, M$_+$), the skyrmion with $Q = -1$ in the top layer generates real-space Berry curvature and thus an equivalent magnetic field $\mathbf{B}^e$ that points along $-z$-direction acting on conduction electrons. As a result, the spin-up conduction electrons are deflected from the current direction and accumulate on the right side of top layer, as shown in top-left panel of Fig. 1c. When transforming into (P$_+$, M$_-$) via reversing magnetic field, the $Q$ of skyrmion is changed from $-1$ to 1, and the signs of real-space Berry curvature and $\mathbf{B}^e$ are also reversed. Consequently, the spin-down conduction electrons are deflected to the left side of top layer; see bottom-left panel of Fig. 1c. Meanwhile, upon switching into (P$_-$, M$_+$)/(P$_-$, M$_-$) through ferroelectricity, as the skyrmions with $Q = -1/1$ are present only in the bottom layer, the real-space Berry curvature emerges in this layer, generating $\mathbf{B}^e$ pointing along $-z/z$-direction. This leads to a rightward/leftward deflection of spin-up/spin-down conduction electrons in the bottom layer (right panels of Fig. 1c). Accordingly, the TLHE is achieved in CrInSe$_3$/In$_2$S$_3$/CrInSe$_3$. And the magnetoelectric coupling enables the TLHE to be effectively controllable through ferroelectricity and magnetism. Moreover, as shown in Fig. 3b, the SkX phase in the top layer can be preserved at a magnetic field of up to 1.4 T, which indicates that the TLHE can be

stable within the magnetic field range of 0.8–1.4 T. Notably, while both anomalous Hall effect (AHE) and TLHE may coexist in CrInSe$_3$/In$_2$S$_3$/ CrInSe$_3$, recent experimental techniques have enabled effective separation of their contributions[56]. For instance, one method involves extracting AHE contribution using a step function, while another isolates TLHE contribution by taking the difference in resistivity between upward and downward magnetic field sweeps.

## Discussion

To confirm the TLHE in CrInSe$_3$/In$_2$S$_3$/CrInSe$_3$, we employ a tight-binding model to study the topological Hall conductance induced by spin textures. The Hamiltonian describing CrInSe$_3$/In$_2$S$_3$/CrInSe$_3$ with the interaction between conduction electrons and spin texture can be written as[57]

$$H = \sum_{\langle i,j \rangle_{\text{inter}}} t_1 c_i^\dagger c_j + \sum_{\langle i,j \rangle_{\text{intra}}} t_2 c_i^\dagger c_j + \sum_{\langle i,\alpha \rangle} J \mathbf{s}_\alpha \cdot \boldsymbol{\sigma} c_{i,s}^\dagger c_{is'} \quad (5)$$

Here, $t_1$ denotes the hoping between top and bottom layers, and $t_2$ represents the hopping between NN sites in each layer. $c_i^\dagger (c_j)$ is the two-component (spin-up and spin-down) creation (annihilation) operator at $i$ site, respectively. $\boldsymbol{\sigma}$ is the vector of Pauli matrices. $J$ represents the Hund's coupling strength between the conduction electron spin and background spin texture. To ensure effective coupling between conduction electrons and spin texture, we take $J = 7t_2$ according to the previous work[57,58]. In our model, the real-space Berry physics emerges from the Hund's coupling term $J \mathbf{s}_\alpha \cdot \boldsymbol{\sigma}$, which mediates the interaction between conduction electrons and the noncoplanar spin texture. This coupling induces a real-space Berry curvature as electrons propagate through regions of finite scalar spin chirality. The topological Hall conductance $\sigma_{xy}(E_F)$ at the Fermi Energy $E_F$ involving the integration of Berry curvature $\Omega_n(\mathbf{k})$ over the 2D Brillouin zone is defined as[57,59]

$$\sigma_{xy}(E_F) = \frac{e^2}{h} \frac{1}{2\pi} \sum_n \int \Omega_n(\mathbf{k}) \cdot f(E_{n\mathbf{k}} - E_F) d^2k \quad (6)$$

Here, $\Omega_n(\mathbf{k})$ is determined by the eigenvectors $u_{nk}$ and corresponding eigenvalues $E_{nk}$, expressed as

$$\Omega_n(\mathbf{k}) = i \sum_{m \neq n} \frac{\langle u_{n\mathbf{k}} | \nabla_{\mathbf{k}} \mathcal{M} \mathcal{H}_{\mathbf{k}} | u_{m\mathbf{k}} \rangle \times \langle u_{m\mathbf{k}} | \nabla_{\mathbf{k}} \mathcal{H}_{\mathbf{k}} | u_{n\mathbf{k}} \rangle}{(E_{n\mathbf{k}} - E_{m\mathbf{k}})^2} \quad (7)$$

where $f(E_{n\mathbf{k}} - E_F)$ is the Fermi−Dirac distribution function. $\mathcal{M}$ is the matrix accounting for the alignment of conduction electrons following

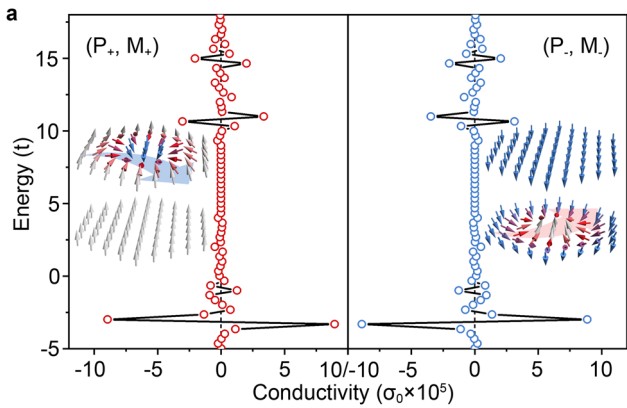

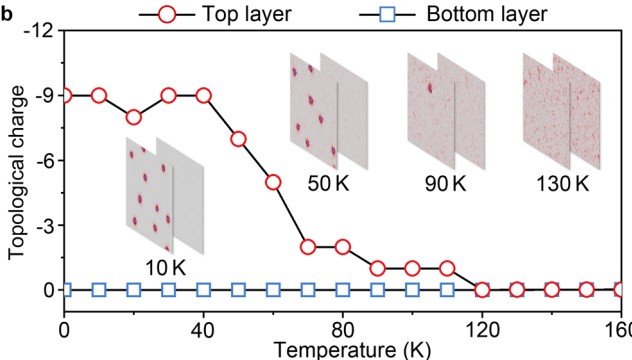

**Fig. 4 | Topological Hall conductivities and temperature effect on spin textures. a** Topological Hall conductivities for $(P_+, M_+)$ and $(P_-, M_-)$ of $CrInSe_3$/$In_2S_3$/$CrInSe_3$. In (**a**), $\sigma_0 = e^2/h$, $t = t_2 = 0.2$ and $t_1 = 5 \times 10^{-4} t_2$ are employed for the Hamiltonian of tight-binding model. **b** Evolutions of topological charge and spin textures of $(P_+, M_+)$ at 0.8 T as functions of the temperature.

the background magnetic skyrmion. Taking $(P_+, M_+)$ and $(P_-, M_-)$ as examples, we present the corresponding $\sigma_{xy}(E_F)$ in Fig. 4a. Here, the spin textures under magnetic field of $\pm 0.8$ T are introduced into the tight-binding model. In detail, the diameter of the skyrmion is approximately $27a_0$ (with $a_0$ being the lattice constant), and the model is thus constructed on a $31 \times 31$ supercell to adequately capture the real-space spin texture. Given the magnetic skyrmions with $Q = -1/1$ of $(P_+, M_+)/(P_-, M_-)$ located in the top/bottom layer, the negative/positive $\sigma_{xy}(E_F)$ localized at the top/bottom layer is obtained when conduction electrons pass through a skyrmion. In each phase, according to the previous works[57,60], there are two factors determining the sigh reversal of $\sigma_{xy}$: one is the band singularities such as a van Hove singularity (at the associated energies the Fermi lines change their character from electron to hole pockets) and the other is band spin splitting (in each spin channel, the electron spin aligns parallel or antiparallel to the spin texture). These phenomena correlate to the hopping strength $t$ and Hund's coupling strength $J$, which are closely linked to the nature of the specific material. The Hamiltonian of $CrInSe_3$/$In_2S_3$/$CrInSe_3$ possesses joint $PT$ symmetry, which ensures the absolute values of $\sigma_{xy}$ being symmetric under the simultaneous reversal of P and M, as shown in Fig. 4a. These results firmly confirm the existence of TLHE in $CrInSe_3$/$In_2S_3$/$CrInSe_3$.

The tight-binding approach presented here serves as a minimal model to demonstrate the fundamental principles of TLHE. While we recognize the importance of material-specific details, explicitly incorporating the full complexity of $CrInSe_3$/$In_2S_3$/$CrInSe_3$ heterostructure—particularly the large-scale skyrmion textures—would be computationally prohibitive. Our simplified s-d model therefore focuses on capturing the essential physics of

electron–skyrmion coupling that underlies the TLHE phenomenon, while maintaining computational tractability for qualitative analysis.

At last, to evaluate the thermal stability of the TLHE, we take $(P_+, M_+)$ under a magnetic field of 0.8 T as an example to investigate the influence of temperature on magnetic skyrmions. Figure 4b illustrates the evolution of topological charge $Q$ with temperature. It can be seen that with increasing temperature from 0 to 40 K, the topological charge of the top layer $Q_{top}$ fluctuates slightly. By further increasing the temperature to 40–110 K, $Q_{top}$ decreases significantly but still approximates an integer, suggesting that the density of magnetic skyrmions decreases with increasing temperature. As shown in Fig. 4b, under the temperature of 0–110 K, the magnetic skyrmions of the top layer gradually diffuse due to the thermal fluctuations, but their overall morphologies are unchanged. Upon increasing the temperature larger than 110 K, $Q_{top}$ drops to zero, signifying the annihilation of magnetic skyrmions within the FM background. For the bottom layer, as shown in Fig. 4b, it remains a trivial FM phase ($Q_{bot} = 0$) as the temperature increases from 0 to 110 K. Therefore, the TLHE in $CrInSe_3$/$In_2S_3$/$CrInSe_3$ can be realized within a relatively wide temperature range of 0–110 K. For practical applications, achieving room-temperature operation of TLHE is crucial. To address this, we propose two complementary strategies: (i) material optimization through the selection of van der Waals magnets with enhanced Curie temperatures, and (ii) strain engineering to amplify magnetic exchange interactions in $CrInSe_3$.

In conclusion, we propose the concept of TLHE in a 2D type-I multiferroic heterostructure and reveal that the TLHE originates from the layer-polarized real-space Berry physics raised from layer-locked magnetic skyrmion with nontrivial topology. This is in sharp contrast to the existing LHEs that are all based on the paradigm of momentum-space Berry phase dependent on fine-tuned bands, as shown in Supplementary Fig. 8. Based on first-principles calculations and atomic spin model simulations, we predict 2D multiferroic heterostructure of $CrInSe_3$/$In_2S_3$/$CrInSe_3$ as the first example of the TLHE. Our analysis reveals two essential criteria for realizing TLHE in $CrInSe_3$/$In_2S_3$/$CrInSe_3$ heterostructure: (i) The FM layers must support intrinsic noncoplanar spin textures (e.g., skyrmions or bimerons); (ii) The FE interlayer must possess stable, switchable out-of-plane polarization with strong coupling to the adjacent magnetic layers. While this study specifically examines $CrInSe_3$/$In_2S_3$/$CrInSe_3$ system, we emphasize that our design principle can be generalized to other FM/FE/FM heterostructures that meet these fundamental requirements.

Finally, we wish to point out that these results are based on freestanding and perfect crystals of $CrInSe_3$/$In_2S_3$/$CrInSe_3$. In practice, substrates and capping layers may introduce strain or doping effects that modify the electronic structure, while crystal defects could lead to localized scattering. Thus, the substrate effects, capping layers, and crystal defects might affect the detailed properties. However, we expect that moderate perturbations would not affect the main results. To aid experimental efforts, we propose that future studies explore defect engineering and substrate selection to mitigate these effects.

## Methods
### First-principles calculations
Our first-principles calculations are conducted using density functional theory as implemented in the Vienna ab initio Simulation Package (VASP)[61,62]. The exchange–correlation interaction is treated with the Perdew–Burke–Ernzerhof functional of generalized gradient approximation (GGA)[63]. The cutoff energy is set to 520 eV. The convergence criteria for the energy and force are set to $10^{-6}$ eV and 0.01 eV/Å, respectively. To account for strong correlation effects, GGA + U method is adopted with $U$=3 eV for $3d$ electrons of Cr

atom[46,64,65]. A Monkhorst–Pack $k$-point mesh of $21 \times 21 \times 1$ is adopted to sample the 2D Brillouin zone. For magnetic interaction parameters calculations, a $1 \times 2\sqrt{3}$ supercell with a $k$-point mesh of $21 \times 6 \times 1$ is applied for exchange interaction parameters, while a $1 \times 4$ supercell with a $k$-point mesh of $16 \times 4 \times 1$ is used for DMI parameters. Further calculation details for magnetic interaction parameters are provided in Supplementary Note 1. Four-state method is employed to verify the accuracy of our calculations[21,66,67]. A vacuum space of 20 Å along the $z$-axis is set to avoid adjacent interactions. DFT-D3 method is utilized to correct the van der Waals interaction[68]. The FE switching pathway is determined using the CI-NEB method[69].

Considering that third-nearest-neighbor (3NN) exchange interactions might play an important role in certain triangular systems[70], we perform test calculations to include both intralayer and interlayer 3NN exchange interactions. The corresponding parameters are listed in Supplementary Table 1. It can be seen that the 3NN terms are relatively weak in $CrInSe_3/In_2S_3/CrInSe_3$ heterostructure. Although these terms prove relatively weak in our heterostructure, we rigorously evaluated their potential impact through atomic spin model simulations. As demonstrated in Supplementary Fig. 9, the inclusion of 3NN interactions preserves both the magnetic skyrmion stability and layer-contrasting magnetism, confirming their minimal influence on the topological magnetism and TLHE.

## Atomic spin model simulations

Atomic spin model simulations are performed using the VAMPIRE package based on the spin Hamiltonian of Eq. (1) and LLG equation[71]:

$$\frac{\partial \mathbf{S}_i}{\partial t} = -\frac{\gamma}{(1+\lambda^2)}\left[\mathbf{S}_i \times \mathbf{B}_{eff}^i + \lambda \mathbf{S}_i \times \left(\mathbf{S}_i \times \mathbf{B}_{eff}^i\right)\right] \quad (8)$$

Here, $\mathbf{S}_i$ is the unit vector of spin moment of site $i$. $\mathbf{B}_{\text{eff}}^i = -\frac{1}{u_s}\frac{\partial H}{\partial \mathbf{S}_i}$ is the on-site effective magnetic field for describing the time evolution of spin structures. $\gamma$ is the gyromagnetic ratio, and $\lambda$ is the damping constant. All simulations are initiated from a high temperature disordered state $(T \gg T_c)$ and gradually cooled to the investigated low temperature with a step of 1 K. The iteration step is set to $10^5$. Stable spin textures are obtained using a $200 \times 200 \times 1$ supercell under periodic boundary conditions, corresponding to a real-space size of approximately $80 \times 80$ nm. In simulations, the $\gamma$ and $\lambda$ are set to $1.76 \times 10^{11}\,T^{-1}s^{-1}$ and 1.0, respectively.

Topological charge $Q$ is computed according to the method proposed by Berg et al. for discrete lattice, which can be expressed as[72]

$$Q = \frac{1}{4\pi}\sum_l qn \quad (9)$$

$$\tan\frac{qn}{2} = \frac{\mathbf{S}_i^n \cdot \left(\mathbf{S}_j^n \times \mathbf{S}_k^n\right)}{1 + \mathbf{S}_i^n \cdot \mathbf{S}_j^n + \mathbf{S}_j^n \cdot \mathbf{S}_k^n + \mathbf{S}_k^n \cdot \mathbf{S}_i^n} \quad (10)$$

Here, $\mathbf{S}_i^n$, $\mathbf{S}_j^n$, $\mathbf{S}_k^n$ are the three spin vectors of the $n^{\text{th}}$ equilateral triangle in the anticlockwise lattice.

## Data availability

The data supporting the findings of this study are available within the article and its Supplementary Information. Additional data are available from the corresponding author upon request.

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

## Acknowledgements

This work is supported by the National Natural Science Foundation of China (grant numbers 12274261 to Y.M., 12074217 to X.L. and U23A20136 to Z.W.). Y.M. acknowledges the support of the Taishan Young Scholar Program of Shandong Province.

## Author contributions

W.D. conceived the research, contributed to calculations, data analysis, and writing. K.D., X.L., Y.D., Z.W., and B.H. contributed to data analysis. Y.M. reviewed and revised the manuscript. Y.M. supervised the project. All authors discussed the results and commented on the manuscript at all stages.

## Competing interests
The authors declare no competing interests.
