## [Transparent Peer Review file · Nature Communications]

Topological Layer Hall Effect in Two-Dimensional Type-I Multiferroic Heterostructure

Corresponding Author: Professor Ying Dai

Version 0:

Reviewer comments:

Reviewer #1

(Remarks to the Author)

This study proposes a topological layer Hall effect (TLHE) originated from the difference in real-space Berry curvature between the upper and lower layers, which extends the concept of the layer Hall effect governed by momentum-space Berry curvature. Although the idea of controlling the creation and annihilation of skyrmions (SkXs) using type-I multiferroic heterostructure has already been proposed in other material systems [K. Huang, et al. Nano Lett. 22, 3349 (2022)], the concept of extending the layer Hall effect to real-space Berry curvature by utilizing the presence/absence of SkXs in each layer is very intriguing. Moreover, since multilayer structures of two-dimensional materials offer a high degree of freedom in their combinations, the proposed concept may overcome the limitations of the layer Hall effect, as the authors claim, and attract interest from researchers in spintronics. Combining first-principles calculations and spin model simulations to demonstrate the topological layer Hall effect is also a standard method for evaluating topological Hall conductivities, however; following concerns should be addressed for the publication.

- (1) The Hamiltonian of the $\text{CrInSe}_3/\text{In}_2\text{S}_3/\text{CrInSe}_3$ system has inversion symmetry to a charge polarization P and a magnetization M . On the other hand, why does the topological Hall conductivity have a difference in energy dependence when P and M are reversed, as shown in Fig. 4(a)?
- (2) Hund's coupling strength J , which determines the coupling strength between a background spin texture and conduction electrons, is crucial to calculate the topological Hall conductivity. However, there is no mention of the specific value of J used in this study or how it was determined.
- (3) There is no comment on the conditions of the spin texture assumed in the calculation of the topological Hall conductivity.
- (4) According to Fig 4(a), the sign of the topological Hall conductivity in the $\text{CrInSe}_3/\text{In}_2\text{S}_3/\text{CrInSe}_3$ system frequently changes as a function of energy, significantly differing from the energy dependence reported in other systems [Göbel, B. et al. Phys. Rev. B 96, 060406(R) (2017). Göbel, B. et al. New J. Phys. 19, 063042 (2017).]. The topological Hall conductivity typically changes its sign at the band (or the density of state) singularities such as a van Hove singularity. Is the energy dependence of the topological Hall conductivity consistent with the DOS in the $\text{CrInSe}_3/\text{In}_2\text{S}_3/\text{CrInSe}_3$ system? Please add a comment on the origin of this oscillatory behavior.
- (5) This research has utilized a freestanding and perfect crystal of $\text{CrInSe}_3/\text{In}_2\text{S}_3/\text{CrInSe}_3$ to demonstrate the topological layer Hall effect. Meanwhile, the system has crystal defects and interactions with the substrate or capping layers in a realistic situation. Could the authors add a comment on these effects on the topological layer Hall effect, which would be highly beneficial for experimental researchers attempting to investigate this phenomenon in the future?

Minor comments

*On page 2, line 42, Ref. 31 should be corrected to Ref. 30.

*In Fig. 2(d), the "(d)" label is missing from the figure.

*On page 7, line 166, The reference to Eq. (6) is incorrect and should be corrected to the appropriate equation. Please verify and update the reference.

*The abbreviation "SOC" appears on page 7 and should be defined at line 179 instead of line 190.

*On page 10, line 267, in Eq. (7), the subscripts in the second term of the numerator are incorrect. It should be corrected from

$(u_n|\dots|u_m)$ to $(u_m|\dots|u_n)$.

Reviewer #2

(Remarks to the Author)

The authors propose a novel theoretical framework for realizing a Topological Layer Hall Effect (TLHE) in a van der Waals heterostructure composed of $\text{CrInSe}_3/\text{In}_2\text{S}_3/\text{CrInSe}_3$. Distinct from conventional Layer Hall Effects that rely on momentum-space Berry curvature, this work attributes TLHE to layer-polarized real-space Berry curvature generated by noncoplanar skyrmion spin textures localized in only one of the magnetic layers. The study leverages first-principles calculations and atomic spin model simulations to demonstrate the layer-selective emergence of skyrmions, controlled via ferroelectric polarization.

The central idea is novel and intellectually stimulating, offering a potentially new direction in spintronics and 2D topological transport. However, several critical points require clarification and further justification before the manuscript can be considered for publication.

1. The authors state that the DMI strengths of the two CrInSe_3 layers become asymmetric depending on the direction of ferroelectric polarization, attributing this behavior to a general "proximity effect." However, this explanation remains qualitative and insufficiently detailed.

- It has been shown that orbital hybridization, local symmetry breaking, and charge redistribution near SOC-active atoms can play crucial roles in modulating the DMI.
- The authors should quantitatively analyze the orbital characteristics, for example via projected density of states (PDOS) or crystal field analysis, to demonstrate how inversion symmetry breaking induced by FE polarization leads to the observed DMI asymmetry.
- If such analysis is not feasible within the current study, at minimum, a more in-depth physical discussion is warranted in the main text.

2. While the proposed TLHE is conceptually distinct from previous LHEs, the manuscript lacks a concrete discussion on the functional benefits or scientific interests of TLHE over conventional Hall effects.

3. The spin model simulations demonstrate the formation of skyrmions in one of the magnetic layers depending on FE polarization and magnetic field. However, important real-space details are missing:
For instance,

- What is the real-space size (in nm) of the simulation cell?
- What is the diameter and density of the skyrmions?

Providing these metrics is important for assessing experimental viability, and for readers intending to reproduce or extend the simulation.

Reviewer #3

(Remarks to the Author)

Referee report: Topological Layer Hall Effect in Two-Dimensional Type-I Multiferroic Heterostructure

The manuscript presents a novel mechanism for realizing a Topological Layer Hall Effect (TLHE) in two-dimensional type-I multiferroic heterostructures, notably demonstrated in a $\text{CrInSe}_3/\text{In}_2\text{S}_3/\text{CrInSe}_3$ system. Unlike previously reported Layer Hall Effects (LHE) based on momentum-space Berry curvature in specific band structures, the proposed TLHE originates from a layer-polarized real-space Berry curvature induced by noncoplanar skyrmionic spin textures. This effect is controllable via the magnetoelectric coupling intrinsic to the heterostructure. The authors support their claims with detailed symmetry analyses, first-principles calculations, tight-binding modeling, and spin dynamics simulations.

The concept of a layer-polarized real-space Berry curvature as a source of Hall-like transport is both novel and significant. It provides a promising direction to bypass the challenges associated with momentum-space Berry curvature in LHEs. The work also offers a new strategy to integrate skyrmion-based phenomena into layertronic applications.

The study is methodologically sound. It combines:

Symmetry analysis to justify the emergence of layer-locked skyrmions.

DFT (with GGA+U) to compute electronic and magnetic properties.

Spin model simulations using realistic parameters to explore skyrmion textures.

Tight-binding modeling for evaluating the Hall conductance.

The calculations are detailed, and key physical mechanisms (e.g., DMI chirality, magnetoelectric coupling, thermal stability) are discussed thoroughly.

Overall, the manuscript is clearly written and logically organized. However, the language would benefit from further polishing, especially in grammar and phrasing (e.g., "picky bands" should be replaced with "fragile bands" or "fine-tuned bands"). Figures are informative, though some of them (e.g., Fig. 1) could be clarified with better labeling and annotations.

The terminology used is consistent with the literature.

Suggestions for Improvement:

Clarify novelty: The distinction between TLHE and prior LHEs should be explicitly reinforced in the introduction and discussion. Consider including a conceptual diagram contrasting real-space vs. momentum-space Berry curvature mechanisms.

Quantitative predictions: While qualitative behavior is compelling, more quantitative estimates (e.g., the magnitude of σ_{xy} in physical units, expected signal strength) would help evaluate experimental feasibility.

Experimental feasibility: Although lattice matching is discussed, further elaboration on the challenges and prospects of fabricating the CrInSe₃/In₂S₃/CrInSe₃ stack would strengthen the practical relevance.

Temperature range: The simulated TLHE stability up to ~110 K is promising but below room temperature. Please discuss possible pathways (material choices or strain engineering) to raise this threshold.

Grammar and language: Careful proofreading is needed to improve language quality. Phrases like “skyrmion physics” and “real-space Berry physics” are overused—consider paraphrasing where appropriate.

The manuscript introduces a compelling and original mechanism that potentially impacts topological spintronics and layertronic systems. Minor revisions to improve clarity and expand on experimental aspects will make it suitable for publication.

Reviewer #4

(Remarks to the Author)

In the manuscript, the authors propose a new concept named as “topological layer Hall effect” (TLHE), which can be caused by the layer-dependent emergence of magnetic skyrmions in multilayer magnetic materials. To be specific, skyrmions can induce topological Hall effect due to the real-space topological nontrivial structure. What is proposed by the authors is that if the skyrmions can emerge in some selected layer, while in other layers the skyrmions diminish, the skyrmion Hall effect could be layer-polarized, which is the main spirit of TLHE. Using first-principles calculations and atomistic spin dynamic simulations, the authors show the possibility for the realization of TLHE in a realistic material platform, CrInSe₃/In₂S₃/CrInSe₃ heterostructure, which is caused by the electric- and magnetic-field-tunable magnetic skyrmions locating on the specific CrInSe₃ layer.

Before I can decide whether to recommend for publishing this manuscript or not, I have several concerns and comments, which should be addressed by the authors.

My main concern about the novelty of this work and the corresponding suggestions: It is a well-known fact that skyrmion can induce topological Hall effect (THE). I think the TLHE is, essentially, another kind of THE effect, thus nothing is new in the physics. In fact, I don't think by mixing the concept of THE and layer-degree of freedom together, one could bring anything profound and tasteful in skyrmion-related physics, unless the authors provide more arguments to me. Moreover, can the author specifically show me in the manuscript that how to utilize the TLHE on possible skyrmion-related information storage as well as spintronics applications, so that the TLHE could bring more possibility on device applications? If the authors have addressed my above concerns, I think this manuscript would be more intuitive and thus closer to the high standard of Nature Communications.

Besides, I also have some minor questions listed below:

- (1) In page 5, the author claimed that the mismatch between CrInSe₃ and In₂S₃ is very small. Can the author show me the lattice constant of CrInSe₃ and In₂S₃, respectively?
- (2) In Fig. 2(c), the authors calculated the atom-resolved SOC energy difference, a detailed description of the method to calculate them is needed, so that others can reproduce these results. Moreover, taking the top part of CrInSe₃ as an example, the authors show that the SOC-induced energy difference on each atom is 22.748, 1.936, -2.249, 0.292, and 1.239, respectively. By summing up them, I find a value of 23.966 meV. What is the relationship of this value (23.966 meV) with the calculated DMI parameter of the top CrInSe₃ (2.006 meV)? Why do they differ so much?
- (3) When calculating the magnetic parameters in triangular systems, the authors need to include the third nearest-neighbouring (3NN) interactions, both intralayer and interlayer, which is proved to be important in triangular systems [Phys. Rev. B 106, 035156 (2022)]. Moreover, the authors should double check their results on magnetic interactions using four-state method [Phys. Rev. Lett. 125, 037203 (2020), Phys. Rev. B 101, 060404(R) (2020), Dalton Trans. 42, 823 (2013)]. Besides, after including the 3NN terms, the authors should reperform LLG simulation to check whether the skyrmions would diminish or not, to check the robustness of their results.
- (4) In page 6, the authors mentioned that “The electric polarizations for P+ and P- phases are calculated to be 3.186 and -3.186 pC/m”. How large the electric field they need to reverse the polarization? Estimate it.
- (5) In the discussion part, the authors show a TB model to evaluate the topological Hall effect. However, I think this TB model is over simplified, as it only considers the s electrons which couples with the skyrmions (s-d model), but the realistic electronic structure of the CrInSe₃/In₂S₃/CrInSe₃ heterostructure is complicated and cannot be simply described by s-d model. Can the author rationalize why it is suitable to use such s-d model?
- (6) In Equation (5), the authors state that the third term represents the Hund's coupling between the electron spin and the spin texture. However, I am curious about the connection between Equations (2)–(4) and Equation (5). Is the value of J taken from Table 1? It would be helpful to include a few words discussing the relationship between these equations. Additionally, considering that magnetic skyrmions can reach nanometer scales, what is the system size represented by the tight-binding model in Equation (5)?
- (7) Is there an anomalous Hall effect in freestanding CrInSe₃? If so, how can one distinguish between the topological Hall conductivity and the anomalous Hall conductivity?
- (8) The paper consistently discusses the influence of real-space Berry physics on carrier transport. What is the real-space Berry physics corresponding to Equation (5) in the tight-binding model?
- (9) Due to the magnetic proximity effect, ferromagnetism can also exist in In₂S₃. The authors should discuss the presence of skyrmions in In₂S₃ and their corresponding Hall conductivities.
- (10) Is the proposed multilayer structure universally applicable? What specific conditions must the ferromagnetic and ferroelectric layers meet to realize the topological layer Hall effect? As shown in Fig. 1, the magnetoelectric coupling in the

CrInSe₃/In₂S₃ system appears to be strong, with the direction of ferroelectric polarization showing a clear correlation with the formation of skyrmions.

Reviewer #5

(Remarks to the Author)

The study highlights sliding ferroelectricity as a potential tool to modulate the magnetic skyrmions. In particular, authors use layer degrees of freedom as a binary index to tune the layer Hall effect, which emerges from the topological spin textures. Currently, magnetoelectric coupling in skyrmionic lattices is of growing interest, and therefore, the current study might merit publication in Nature Communications. However, I have a few suggestions which should be considered before publication.

1) In particular, authors use CrInSe₃/In₂S₃/CrInSe₃ hetero-stacking. I assume In₂Se₃ is the prototype source of ferroelectricity. I believe CrInSe₃/CrInSe₃ bilayer may also result in polar stacking. Is there any specific reason for choosing trilayer stacking?

2) Authors discuss the ab initio results for Neel-type skyrmions. Should the results also be applicable to other topological textures, i.e., merons?

3) The exchange coupling is included up to the second-nearest neighbors, which may sometimes oversimplify the model. Sometimes anisotropic exchange or DMI may also play an important role.

Version 1:

Reviewer comments:

Reviewer #1

(Remarks to the Author)

The authors have addressed my concerns appropriately.

Reviewer #3

(Remarks to the Author)

I believe all the issues raised for the previous version have been addressed. Therefore I recommend the paper to be published in its current form.

Reviewer #4

(Remarks to the Author)

The revised manuscript has answered all my questions and I believe it is suitable for publication.

Reviewer #5

(Remarks to the Author)

Authors have satisfactorily addressed my concerns. I think the manuscript might merit publication in Nature Communications.

Response Letter to Reviewers

For Reviewer #1

This study proposes a topological layer Hall effect (TLHE) originated from the difference in real-space Berry curvature between the upper and lower layers, which extends the concept of the layer Hall effect governed by momentum-space Berry curvature. Although the idea of controlling the creation and annihilation of skyrmions (SkXs) using type-I multiferroic heterostructure has already been proposed in other material systems [K. Huang, et al. Nano Lett. 22, 3349 (2022)], **the concept of extending the layer Hall effect to real-space Berry curvature by utilizing the presence/absence of SkXs in each layer is very intriguing.** Moreover, since multilayer structures of two-dimensional materials offer a high degree of freedom in their combinations, **the proposed concept may overcome the limitations of the layer Hall effect, as the authors claim, and attract interest from researchers in spintronics.** Combining first-principles calculations and spin model simulations to demonstrate the topological layer Hall effect is also a standard method for evaluating topological Hall conductivities, however; following concerns should be addressed for the publication.

We appreciate the reviewer's high evaluation on this work. His/her comments are closely followed.

Comment 1: The Hamiltonian of the CrInSe₃/In₂S₃/CrInSe₃ system has inversion symmetry to a charge polarization P and a magnetization M. On the other hand, why does the topological Hall conductivity have a difference in energy dependence when P and M are reversed, as shown in Fig. 4(a)?

Response 1: Thanks for this comment. In fact, the slight difference in energy dependence of σ_{xy} between (P+, M+) and (P-, M-) phases in Fig. 4(a) arises from small deviations in the simulated spin textures. Closely following the reviewer's comments, we performed additional calculations and included the following sentence on page 12: "The Hamiltonian of CrInSe₃/In₂S₃/CrInSe₃ possesses joint *PT* symmetry, which ensure the absolute values of σ_{xy} being symmetric under the simultaneous reversal of P and M, as shown in Fig. 4(a)." And Fig. 4(a) was revised.

Comment 2: Hund's coupling strength *J*, which determines the coupling strength between a background spin texture and conduction electrons, is crucial to calculate the topological Hall conductivity. However, there is no mention of the specific value of *J* used in this study or how it was determined.

Response 2: Closely following the reviewer's comment, we included the following sentence on page 11: "To ensure effective coupling between conduction electrons and spin texture, we take $J = 7t_2$ according to the previous work [57,58]." Refs. [57,58] were added.

Comment 3: There is no comment on the conditions of the spin texture assumed in the calculation of the topological Hall conductivity.

Response 3: Closely following the reviewer's comment, we revised the following sentences on page 11: "Taking (P+, M+) and (P-, M-) as examples, we present the corresponding $\sigma_{xy}(E_F)$ in Fig. 4(a). Here, the spin textures under magnetic field of ± 0.8 T are introduced into the tight-binding model."

Comment 4: According to Fig 4(a), the sign of the topological Hall conductivity in the CrInSe₃/In₂S₃/CrInSe₃ system frequently changes as a function of energy, significantly differing from the energy dependence reported in other systems [Göbel, B. et al. Phys. Rev. B 96, 060406(R) (2017). Göbel, B. et al. New J. Phys. 19, 063042 (2017)]. The topological Hall conductivity typically changes its sign at the band (or the density of state) singularities such as a van Hove singularity. Is the energy dependence of the topological Hall conductivity consistent with the DOS in the CrInSe₃/In₂S₃/CrInSe₃ system? Please add a comment on the origin of this oscillatory behavior.

Response 4: We acknowledge that the apparent frequent oscillatory behavior in Fig. 4(a) results from an error in our energy-axis scaling during data processing. Fig. 4(a) was revised in the revised manuscript. Closely following the reviewer's comment, we included the following sentences on page 11: "In each phase, according to the previous works [57,60], there are two factors determining the sign reversal of σ_{xy} : one is the band singularities such as a van Hove singularity (at the associated energies the Fermi lines change their character from electron to hole pockets) and the other is band spin splitting (in each spin channel, the electron spin aligns parallel or antiparallel to the spin texture). These phenomena correlate to the hopping strength t and Hund's coupling strength J , which are closely linked to the nature of specific material." Refs. [57,60] were added.

Regarding calculating the DOS of such spin structure, frankly speaking, it is really out of our ability to do it at present. We are really sorry for that.

Comment 5: This research has utilized a freestanding and perfect crystal of CrInSe₃/In₂S₃/CrInSe₃ to demonstrate the topological layer Hall effect. Meanwhile, the system has crystal defects and interactions with the substrate or capping layers in a realistic situation. Could the authors add a comment on these effects on the topological layer Hall effect, which would be highly beneficial for experimental researchers attempting to investigate this phenomenon in the future?

Response 5: Closely following the reviewer's comment, we included the following sentences on page 13: "Finally, we wish to point out that these results are based on freestanding and perfect crystal of CrInSe₃/In₂S₃/CrInSe₃. In practice, substrates and capping layers may introduce strain or doping effects that modify the electronic structure, while crystal defects could lead to localized scattering. Thus, the substrate effects, capping layers, and crystal defects might affect the detailed properties. However, we expect that moderate perturbations would not affect the main results. To aid experimental efforts, we propose that future studies explore defect engineering and substrate selection to mitigate these effects."

Minor comments 1: On page 2, line 42, Ref. 31 should be corrected to Ref. 30.

Response 1: Sorry for the typo, we have corrected it in the revised manuscript.

Minor comments 2: In Fig. 2(d), the "(d)" label is missing from the figure.

Response 2: Sorry for the typo, we have corrected it in the revised manuscript.

Minor comments 3: On page 7, line 166, The reference to Eq. (6) is incorrect and should be corrected

to the appropriate equation. Please verify and update the reference.

Response 3: Sorry for the typo, we have corrected it in the revised manuscript.

Minor comments 4: The abbreviation "SOC" appears on page 7 and should be defined at line 179 instead of line 190.

Response 4: Sorry for the typo, we have corrected it in the revised manuscript.

Minor comments 5: On page 10, line 267, in Eq. (7), the subscripts in the second term of the numerator are incorrect. It should be corrected from $\langle u_n | \dots | u_m \rangle$ to $\langle u_m | \dots | u_n \rangle$.

Response 5: Sorry for the typo, we have corrected it in the revised manuscript.

For Reviewer #2

The authors propose a novel theoretical framework for realizing a Topological Layer Hall Effect (TLHE) in a van der Waals heterostructure composed of CrInSe₃/In₂S₃/CrInSe₃. Distinct from conventional Layer Hall Effects that rely on momentum-space Berry curvature, this work attributes TLHE to layer-polarized real-space Berry curvature generated by noncoplanar skyrmion spin textures localized in only one of the magnetic layers. The study leverages first-principles calculations and atomic spin model simulations to demonstrate the layer-selective emergence of skyrmions, controlled via ferroelectric polarization.

The central idea is novel and intellectually stimulating, offering a potentially new direction in spintronics and 2D topological transport. However, several critical points require clarification and further justification before the manuscript can be considered for publication.

We appreciate the reviewer's positive evaluation on this work. His/her comments are closely followed.

Comment 1: The authors state that the DMI strengths of the two CrInSe₃ layers become asymmetric depending on the direction of ferroelectric polarization, attributing this behavior to a general "proximity effect." However, this explanation remains qualitative and insufficiently detailed.

- It has been shown that orbital hybridization, local symmetry breaking, and charge redistribution near SOC-active atoms can play crucial roles in modulating the DMI.
- The authors should quantitatively analyze the orbital characteristics, for example via projected density of states (PDOS) or crystal field analysis, to demonstrate how inversion symmetry breaking induced by FE polarization leads to the observed DMI asymmetry.
- If such analysis is not feasible within the current study, at minimum, a more in-depth physical discussion is warranted in the main text.

Response 1: Closely following the reviewer's comments, we revised the following sentences on page 8: "The asymmetric nature of DMI and single ion anisotropy can be attributed to the different proximity effects exerted by α -In₂S₃. As illustrated in **Fig. S2**, the charge transfers from bottom CrInSe₃ layer to In₂S₃ layer, resulting in charge redistributions around Cr atoms and SOC-active Se atoms in bottom CrInSe₃ layer. In contrast, no such charge transfer is observed between top CrInSe₃ layer and In₂S₃ layer, indicating a much weaker proximity effect. This leads to distinct electronic

environments near the Fermi level for the top and bottom CrInSe₃ layers, particularly in terms of orbital occupation (see **Fig. S6**). It should be noted that both DMI and single ion anisotropy originate from SOC-mediated interactions between occupied and unoccupied states near the Fermi level [54,55]. Consequently, the SOC-mediated interaction between occupied and unoccupied states becomes layer-dependent, giving rise to the observed asymmetry in both DMI and single ion anisotropy between the two CrInSe₃ layers.” Fig. S6 was included in the Supplementary Information. Refs. [54,55] were added.

Comment 2: While the proposed TLHE is conceptually distinct from previous LHEs, the manuscript lacks a concrete discussion on the functional benefits or scientific interests of TLHE over conventional Hall effects.

Response 2: Closely following the reviewer’s comment, we included the following sentences on page 4: “Meanwhile, TLHE introduces a layer degree of freedom as a tunable parameter for transport, which distinguishes it from conventional Hall effects (e.g., quantum anomalous Hall effect [43] and anomalous valley Hall effect [44]). And as compared with LHE, TLHE represents an important step towards spatial engineering of topological spintronics.” Refs. [43,44] were added.

Comment 3: The spin model simulations demonstrate the formation of skyrmions in one of the magnetic layers depending on FE polarization and magnetic field. However, important real-space details are missing: For instance,

- What is the real-space size (in nm) of the simulation cell?
- What is the diameter and density of the skyrmions?

Providing these metrics is important for assessing experimental viability, and for readers intending to reproduce or extend the simulation.

Response 3: Closely following the reviewer’s comments, we made the following revisions.

- a) On page 15, we revised the following sentence: “Stable spin textures are obtained using a $200 \times 200 \times 1$ supercell under periodic boundary conditions, corresponding to a real-space size of approximately 80×80 nm.”
- b) On page 9, we revised the following sentence: “Meanwhile, as shown in **Figs. 3(a)** and **S7**, the skyrmion sizes in the two layers are significantly distinct, which is attributed to the differences in strength of magnetic interactions.” Fig. S7 was included in the Supplementary Information.
- c) On page 9, we revised the following sentences: “It is evident that the density of skyrmion in top and bottom layers exhibits different responses to magnetic field. Specifically, in top layer, as magnetic field increases from 0 T to 0.8 T, it gradually increases and reaches a maximum of 1.5×10^{-3} per nm² (9 per supercell) at 0.8 T. While in bottom layer, it first increases under magnetic field of 0 - 0.3 T, but with further increasing magnetic field, it decreases and eventually drops to zero at 0.8 T.”

For Reviewer #3

The manuscript presents a novel mechanism for realizing a Topological Layer Hall Effect (TLHE) in two-dimensional type-I multiferroic heterostructures, notably demonstrated in a

CrInSe₃/In₂S₃/CrInSe₃ system. Unlike previously reported Layer Hall Effects (LHE) based on momentum-space Berry curvature in specific band structures, the proposed TLHE originates from a layer-polarized real-space Berry curvature induced by noncoplanar skyrmionic spin textures. This effect is controllable via the magnetoelectric coupling intrinsic to the heterostructure. The authors support their claims with detailed symmetry analyses, first-principles calculations, tight-binding modeling, and spin dynamics simulations.

The concept of a layer-polarized real-space Berry curvature as a source of Hall-like transport is **both novel and significant. It provides a promising direction to bypass the challenges associated with momentum-space Berry curvature in LHEs. The work also offers a new strategy to integrate skyrmion-based phenomena into layertronic applications.**

The study is methodologically sound. It combines:

Symmetry analysis to justify the emergence of layer-locked skyrmions.

DFT (with GGA+U) to compute electronic and magnetic properties.

Spin model simulations using realistic parameters to explore skyrmion textures.

Tight-binding modeling for evaluating the Hall conductance.

The calculations are detailed, and key physical mechanisms (e.g., DMI chirality, magnetoelectric coupling, thermal stability) are discussed thoroughly.

Overall, the manuscript is clearly written and logically organized. However, the language would benefit from further polishing, especially in grammar and phrasing (e.g., “picky bands” should be replaced with “fragile bands” or “fine-tuned bands”). Figures are informative, though some of them (e.g., Fig. 1) could be clarified with better labeling and annotations. The terminology used is consistent with the literature.

The manuscript introduces a compelling and original mechanism that potentially impacts topological spintronics and layertronic systems. Minor revisions to improve clarity and expand on experimental aspects will make it suitable for publication.

We appreciate the reviewer’s positive evaluation on this work. His/her comments are closely followed.

Comment 1: Clarify novelty: The distinction between TLHE and prior LHEs should be explicitly reinforced in the introduction and discussion. Consider including a conceptual diagram contrasting real-space vs. momentum-space Berry curvature mechanisms.

Response 1: Closely following the reviewer’s comments, we made the following revisions.

a) On page 2, we revised the following sentences: “Via symmetry and model analysis, we reveal that TLHE originates from the layer-polarized real-space Berry physics, which arises from layer-locked magnetic skyrmion with nontrivial topology. This mechanism is fundamentally distinct from the existing LHEs that are all based on the paradigm of momentum-space Berry phase depended on fine-tuned bands.”

b) On page 13, we revised the following sentences: “In conclusion, we propose the concept of TLHE in two-dimensional type-I multiferroic heterostructure, and reveal that the TLHE originates from

the layer-polarized real-space Berry physics raised from layer-locked magnetic skyrmion with nontrivial topology. This is in sharp contrast to the existing LHEs that are all based on the paradigm of momentum-space Berry phase depended on fine-tuned bands, as shown in **Fig. S8**.” Fig. S8 was included in the Supplementary Information.

c) On page 4, we included the following sentence: “And as compared with LHE, TLHE represents an important step towards spatial engineering of topological spintronics.”

Comment 2: Quantitative predictions: While qualitative behavior is compelling, more quantitative estimates (e.g., the magnitude of σ_{xy} in physical units, expected signal strength) would help evaluate experimental feasibility.

Response 2: Thanks for this comment. We agree that quantitative estimates of the topological Hall conductivity would be valuable for assessing experimental feasibility. However, we would like to emphasize that our current study is based on a tight-binding model, which is intended to qualitatively demonstrate the possibility of realizing the topological layer Hall effect. Due to the simplified nature of the model, providing quantitative predictions for σ_{xy} in physical units is challenging at this stage. Nevertheless, we hope that our qualitative insights can offer meaningful guidance for future theoretical and experimental investigations on real material systems.

To reflect this point, we included the following sentences on page 12: “The tight-binding approach presented here serves as a minimal model to demonstrate the fundamental principles of TLHE. While we recognize the importance of material-specific details, explicitly incorporating the full complexity of CrInSe₃/In₂S₃/CrInSe₃ heterostructure—particularly the large-scale skyrmion textures—would be computationally prohibitive. Our simplified s-d model therefore focuses on capturing the essential physics of electron-skyrmion coupling that underlies the TLHE phenomenon, while maintaining computational tractability for qualitative analysis.”

Comment 3: Experimental feasibility: Although lattice matching is discussed, further elaboration on the challenges and prospects of fabricating the CrInSe₃/In₂S₃/CrInSe₃ stack would strengthen the practical relevance.

Response 3: Closely following the reviewer’s comment, we revised the following sentences on page 6: “Given this small lattice mismatch and the van der Waals layered nature of both CrInSe₃ and In₂S₃, the fabrication of CrInSe₃/In₂S₃/CrInSe₃ heterostructure is practically feasible. Recent advances in mechanical exfoliation, van der Waals stacking, and epitaxial growth techniques such as chemical vapor deposition and molecular beam epitaxy provide promising pathways for realizing such layered structures experimentally [48].” Ref. [48] was added.

Comment 4: Temperature range: The simulated TLHE stability up to ~110 K is promising but below room temperature. Please discuss possible pathways (material choices or strain engineering) to raise this threshold.

Response 4: Closely following the reviewer’s comment, we included the following sentences on page 13: “For practical applications, achieving room-temperature operation of TLHE is crucial. To

address this, we propose two complementary strategies: (i) material optimization through the selection of van der Waals magnets with enhanced Curie temperatures, and (ii) strain engineering to amplify magnetic exchange interactions in CrInSe₃.”

Comment 5: Grammar and language: Careful proofreading is needed to improve language quality. Phrases like “skyrmion physics” and “real-space Berry physics” are overused—consider paraphrasing where appropriate.

Response 5: Thanks for this comment. We have carefully revised the manuscript to improve the overall language quality. In particular, we rephrased repetitive expressions such as “skyrmion physics” and “real-space Berry physics” where appropriate to enhance clarity and avoid redundancy.

Comment 6: The language would benefit from further polishing, especially in grammar and phrasing (e.g., “picky bands” should be replaced with “fragile bands” or “fine-tuned bands”).

Response 6: Closely following the reviewer’s comment, we carefully revised the manuscript to improve grammar and phrasing throughout. The term “picky bands” was replaced with “fine-tuned bands”.

For Reviewer #4

In the manuscript, the authors propose a new concept named as “topological layer Hall effect” (TLHE), which can be caused by the layer-dependent emergence of magnetic skyrmions in multilayer magnetic materials. To be specific, skyrmions can induce topological Hall effect due to the real-space topological nontrivial structure. What is proposed by the authors is that if the skyrmions can emerge in some selected layer, while in other layers the skyrmions diminish, the skyrmion Hall effect could be layer-polarized, which is the main spirit of TLHE. Using first-principles calculations and atomistic spin dynamic simulations, the authors show the possibility for the realization of TLHE in a realistic material platform, CrInSe₃/In₂S₃/CrInSe₃ heterostructure, which is caused by the electric- and magnetic-field-tunable magnetic skyrmions locating on the specific CrInSe₃ layer.

Before I can decide whether to recommend for publishing this manuscript or not, I have several concerns and comments, which should be addressed by the authors.

My main concern about the novelty of this work and the corresponding suggestions: It is a well-known fact that skyrmion can induce topological Hall effect (THE). I think the TLHE is, essentially, another kind of THE effect, thus nothing is new in the physics. In fact, I don’t think by mixing the concept of THE and layer-degree of freedom together, one could bring anything profound and tasteful in skyrmion-related physics, unless the authors provide more arguments to me. Moreover, can the author specifically show me in the manuscript that how to utilize the TLHE on possible skyrmion-related information storage as well as spintronics applications, so that the TLHE could bring more possibility on device applications? If the authors have addressed my above concerns, I think this manuscript would be more intuitive and thus closer to the high standard of Nature Communications.

We sincerely appreciate the reviewer's thoughtful evaluation of our work. We would like to highlight

that electrons possess not only charge, spin, and valley degrees of freedom, but also a layer degree of freedom, which gives rise to the novel layer Hall effect (LHE). First proposed in Nature 595, 521 (2021), LHE has emerged as an important new member of the Hall effect family, holding significant fundamental and applied potential in condensed matter physics. In this paradigm, the layer degree of freedom acts as a pseudospin, analogous to spin in ferromagnetism, opening exciting possibilities for future electronic and spintronic applications.

However, all existing studies on LHE are rooted in the paradigm of momentum-space Berry phase, which necessarily requires for bands that are not easily to be satisfied. This rises an outstanding challenge for the development of LHE. In this work, we made a key breakthrough by proposing a mechanism that couples magnetic skyrmion and layer physics in two-dimensional type-I multiferroic heterostructure, generating the concept of TLHE. Distinct from the existing LHEs that are all driven by momentum-space Berry phase relied on picky bands, TLHE correlates to the layer-polarized real-space Berry physics from noncoplanar spin textures of layer-locked magnetic skyrmion with nontrivial topology. These findings overcome the restrictions of LHE.

Furthermore, by combining the LHE with layer degree of freedom control, TLHE provides a powerful new approach to manipulate topological transport properties—a capability that is highly desirable for advancing topological spintronics applications.

Closely following the reviewer's comments, we revised the corresponding sentences to reflect these points.

Comment 1: In page 5, the author claimed that the mismatch between CrInSe₃ and In₂S₃ is very small. Can the author show me the lattice constant of CrInSe₃ and In₂S₃, respectively?

Response 1: Closely following the reviewer's comment, we included the following sentence on page 6: "The lattice constants of CrInSe₃ and α -In₂S₃ are optimized to be 3.92 and 3.94 Å, respectively, yielding a lattice mismatch of only 0.7%."

Comment 2: In Fig. 2(c), the authors calculated the atom-resolved SOC energy difference, a detailed description of the method to calculate them is needed, so that others can reproduce these results. Moreover, taking the top part of CrInSe₃ as an example, the authors show that the SOC-induced energy difference on each atom is 22.748, 1.936, -2.249, 0.292, and 1.239, respectively. By summing up them, I find a value of 23.966 meV. What is the relationship of this value (23.966 meV) with the calculated DMI parameter of the top CrInSe₃ (2.006 meV)? Why do they differ so much?

Response 2: Thanks for this comment. To calculate DMI parameters, we constructed a 1×4 supercell and considered the left- and right-hand spin-spiral configurations, as detailed in **Fig. S4**. Based on the analysis of the contributions of DMI in the two spin configurations, the d_{\parallel} can be obtained by $d_{\parallel} = \frac{\Delta E}{12}$, where ΔE is energy difference between two spin configurations. The detailed calculation method was provided in the Supplementary Information. As for atomic-layer-resolved SOC energy difference (ΔE_{soc}), it is the energy projected onto each atom layer by ΔE . Therefore, there is an

approximately 12-fold relationship between the sum of ΔE_{soc} and $d_{||}$.

To avoid the potential misleading, we revised the following sentences on page 8: “To further investigate the origin of the large DMI in P+ phase, we examined the atomic-layer-resolved SOC energy difference ΔE_{soc} , defined as the energy difference between right- and left-hand spin-spiral configurations, as shown in **Fig. S5**.” Fig. S5 was included in the Supplementary Information.

And we revised the following sentence on page 14: “a 1×4 supercell...is used for DMI parameters”.

Comment 3: When calculating the magnetic parameters in triangular systems, the authors need to include the third nearest-neighbouring (3NN) interactions, both intralayer and interlayer, which is proved to be important in triangular systems [Phys. Rev. B 106, 035156 (2022)]. Moreover, the authors should double check their results on magnetic interactions using four-state method [Phys. Rev. Lett. 125, 037203 (2020), Phys. Rev. B 101, 060404(R) (2020), Dalton Trans. 42, 823 (2013)]. Besides, after including the 3NN terms, the authors should reperform LLG simulation to check whether the skyrmions would diminish or not, to check the robustness of their results.

Response 3: Closely following the reviewer’s comments, we performed additional calculations to include both intralayer and interlayer 3NN exchange interactions. On page 14, we included the following sentences: “Considering that third-nearest-neighbor (3NN) exchange interactions might play an important role in certain triangular systems [70], we perform test calculations to include both intralayer and interlayer 3NN exchange interactions. The corresponding parameters are listed in **Table S1**. It can be seen that the 3NN terms are relatively weak in CrInSe₃/In₂S₃/CrInSe₃ heterostructure. Although these terms prove relatively weak in our heterostructure, we rigorously evaluated their potential impact through atomic spin model simulations. As demonstrated in **Fig. S9**, the inclusion of 3NN interactions preserves both the magnetic skyrmion stability and layer-contrasting magnetism, confirming their minimal influence on the topological magnetism and TLHE.” Ref. [70] was added. Table S1 and Fig. S9 were included in the Supplementary Information.

Regarding the four-state method, while the large size of our heterostructure makes the full application of this method computationally demanding, we have carried out a benchmark calculation using the four-state method by taking the intralayer nearest-neighbor exchange interactions J_1^{top} and J_1^{bot} as representative examples. The results ($J_1^{top} = 24.97$ meV and $J_1^{bot} = 24.27$ meV) are in close agreement with those derived from the method used in our study, confirming the accuracy of our calculations. Closely following the reviewer’s comment, we included the following sentence on page 14: “Four-state method is employed to verify the accuracy of our calculations [21,66,67].” Refs. [21,66,67] were added.

Comment 4: In page 6, the authors mentioned that “The electric polarizations for P+ and P- phases are calculated to be 3.186 and -3.186 pC/m”. How large the electric field they need to reverse the polarization? Estimate it.

Response 4: Closely following the reviewer’s comment, we included the following sentences on page 6: “For comparison, the ferroelectric transition barrier of monolayer In_2Se_3 has been theoretically estimated to be 66 meV [49], and the corresponding experimental electric field required for polarization switching is about 200 kV/cm [50]. Based on this fact, the electric field needed to reverse the polarization of In_2S_3 in the $\text{CrInSe}_3/\text{In}_2\text{S}_3/\text{CrInSe}_3$ heterostructure is expected to be below 200 kV/cm, confirming the feasibility of realizing ferroelectricity in this system.” Refs. [49,50] were added.

Comment 5: In the discussion part, the authors show a TB model to evaluate the topological Hall effect. However, I think this TB model is over simplified, as it only considers the s electrons which couples with the skyrmions (s-d model), but the realistic electronic structure of the $\text{CrInSe}_3/\text{In}_2\text{S}_3/\text{CrInSe}_3$ heterostructure is complicated and cannot be simply described by s-d model. Can the author rationalize why it is suitable to use such s-d model?

Response 5: Closely following the reviewer’s comment, we included the following sentences on page 12: “The tight-binding approach presented here serves as a minimal model to demonstrate the fundamental principles of TLHE. While we recognize the importance of material-specific details, explicitly incorporating the full complexity of $\text{CrInSe}_3/\text{In}_2\text{S}_3/\text{CrInSe}_3$ heterostructure—particularly the large-scale skyrmion textures—would be computationally prohibitive. Our simplified s-d model therefore focuses on capturing the essential physics of electron-skyrmion coupling that underlies the TLHE phenomenon, while maintaining computational tractability for qualitative analysis.”

Comment 6: In Equation (5), the authors state that the third term represents the Hund's coupling between the electron spin and the spin texture. However, I am curious about the connection between Equations (2)–(4) and Equation (5). Is the value of J taken from Table 1? It would be helpful to include a few words discussing the relationship between these equations. Additionally, considering that magnetic skyrmions can reach nanometer scales, what is the system size represented by the tight-binding model in Equation (5)?

Response 6: Closely following the reviewer’s comments, we made the following revisions.

a) On page 11, we included the following sentence: “To ensure effective coupling between conduction electrons and spin texture, we take $J = 7t_2$ according to the previous work [57,58].” Refs. [57,58] were added.

b) On page 11, we revised the following sentences: “Taking (P_+, M_+) and (P_-, M_-) as examples, we present the corresponding $\sigma_{xy}(E_F)$ in Fig. 4(a). Here, the spin textures under magnetic field of ± 0.8 T are introduced into the tight-binding model. In detail, the diameter of skyrmion is approximately $27a_0$ (with a_0 being the lattice constant), and the model is thus constructed on a 31×31 supercell to adequately capture the real-space spin texture.”

Comment 7: Is there an anomalous Hall effect in freestanding CrInSe_3 ? If so, how can one distinguish between the topological Hall conductivity and the anomalous Hall conductivity?

Response 7: Closely following the reviewer’s comments, we included the following sentences on

page 10: “Notably, while both anomalous Hall effect (AHE) and TLHE may coexist in CrInSe₃/In₂S₃/CrInSe₃, recent experimental techniques have enabled effective separation of their contributions [56]. For instance, one method involves extracting AHE contribution using a step function, while another isolates TLHE contribution by taking the difference in resistivity between upward and downward magnetic field sweeps.” Ref. [56] was added.

Comment 8: The paper consistently discusses the influence of real-space Berry physics on carrier transport. What is the real-space Berry physics corresponding to Equation (5) in the tight-binding model?

Response 8: Closely following the reviewer’s comment, we included the following sentences on page 11: “In our model, the real-space Berry physics emerges from the Hund's coupling term $J\mathbf{s}_\alpha \cdot \boldsymbol{\sigma}$, which mediates the interaction between conduction electrons and the noncoplanar spin texture. This coupling induces a real-space Berry curvature as electrons propagate through regions of finite scalar spin chirality.”

Comment 9: Due to the magnetic proximity effect, ferromagnetism can also exist in In₂S₃. The authors should discuss the presence of skyrmions in In₂S₃ and their corresponding Hall conductivities.

Response 9: Closely following the reviewer’s comment, we included the following sentences on page 6: “While magnetic proximity effects could theoretically induce weak magnetic moments in the In₂S₃ layer, our calculations show these moments are negligible ($< 0.001 \mu_B/\text{atom}$) compared to those in Cr atom. This three-orders-of-magnitude difference confirms that In₂S₃ cannot support topological spin textures. Therefore, our analysis focuses exclusively on skyrmion formation in the CrInSe₃ layers.”

Comment 10: Is the proposed multilayer structure universally applicable? What specific conditions must the ferromagnetic and ferroelectric layers meet to realize the topological layer Hall effect? As shown in Fig. 1, the magnetoelectric coupling in the CrInSe₃/In₂S₃ system appears to be strong, with the direction of ferroelectric polarization showing a clear correlation with the formation of skyrmions.

Response 10: Closely following the reviewer’s comments, we included the following sentences on page 13: “Our analysis reveals two essential criteria for realizing TLHE in CrInSe₃/In₂S₃/CrInSe₃ heterostructure: (i) The ferromagnetic layers must support intrinsic noncoplanar spin textures (e.g., skyrmions or bimerons); (ii) The ferroelectric interlayer must possess stable, switchable out-of-plane polarization with strong coupling to the adjacent magnetic layers. While this study specifically examines CrInSe₃/In₂S₃/CrInSe₃ system, we emphasize that our design principle can be generalized to other FM/FE/FM heterostructures that meet these fundamental requirements.”

For Reviewer #5

The study highlights sliding ferroelectricity as a potential tool to modulate the magnetic skyrmions. In particular, authors use layer degrees of freedom as a binary index to tune the layer Hall effect, which emerges from the topological spin textures. **Currently, magnetoelectric coupling in**

skyrmionic lattices is of growing interest, and therefore, the current study might merit publication in Nature Communications. However, I have a few suggestions which should be considered before publication.

We appreciate the reviewer's positive evaluation on this work. His/her comments are closely followed.

Comment 1: In particular, authors use CrInSe₃/In₂S₃/CrInSe₃ hetero-stacking. I assume In₂Se₃ is the prototype source of ferroelectricity. I believe CrInSe₃/CrInSe₃ bilayer may also result in polar stacking. Is there any specific reason for choosing trilayer stacking?

Response 1: Thanks for this comment. The hallmark of TLHE lies in its electrically switchable layer polarization. In the case of CrInSe₃, it exhibits two non-degenerate polarization states with distinct magnetic properties, and features high ferroelectric switching barrier [Appl. Phys. Lett. 124, 162902 (2024)], which renders them unsuitable for realizing reversible TLHE.

Additionally, we revised the following sentence on page 3: "Since our target is to generate the coupling between magnetic skyrmion and layer physics, the systems studied here should be multilayers with layer polarization and time-reversal symmetry breaking. We also require the layer polarized physics to be reversible. These requirements naturally point to 2D multiferroic heterostructure. Inspired by these insights, we here focus on 2D type-I multiferroic heterostructure with sandwiching one FE single-layer between two ferromagnetic (FM) single-layers (referred to as A-layer and B-layer) with large Dzyaloshinskii-Moriya interaction (DMI)."

Comment 2: Authors discuss the ab initio results for Neel-type skyrmions. Should the results also be applicable to other topological textures, i.e., merons?

Response 2: Closely following the reviewer's comment, we included the following sentences on page 3: "In this regard, different topological spin textures [e.g., skyrmion (SkX) and bimeron] might be expected in these two FM layers. Without loss of generality, as illustrated in top-left panel of Fig. 1(c), we assume that A-layer favors SkX phase, while B-layer presents a trivial FM phase [referred to as (P₊, M₊) phase]."

Comment 3: The exchange coupling is included up to the second-nearest neighbors, which may sometimes oversimplify the model. Sometimes anisotropic exchange or DMI may also play an important role.

Response 3: Closely following the reviewer's comment, we performed additional calculations. We find that the anisotropic exchange interaction in CrInSe₃ is only -0.006 meV, and the second-nearest-neighbor DMI is only 0.002 meV. These negligible values justify their exclusion from our main calculations. We have added this clarification on page 3 of Supplementary Information: "Test calculations confirm that both the anisotropic symmetric exchange (-0.006 meV) and second-nearest-neighbor DMI (0.002 meV) in CrInSe₃ are exceptionally weak, validating their exclusion from our spin Hamiltonian."

Moreover, we also performed additional calculations to include both intralayer and interlayer 3NN exchange interactions. On page 14, we included the following sentences: "Considering that

third-nearest-neighbor (3NN) exchange interactions might play an important role in certain triangular systems [70], we perform test calculations to include both intralayer and interlayer 3NN exchange interactions. The corresponding parameters are listed in **Table S1**. It can be seen that the 3NN terms are relatively weak in CrInSe₃/In₂S₃/CrInSe₃ heterostructure. Although these terms prove relatively weak in our heterostructure, we rigorously evaluated their potential impact through atomic spin model simulations. As demonstrated in **Fig. S9**, the inclusion of 3NN interactions preserves both the magnetic skyrmion stability and layer-contrasting magnetism, confirming their minimal influence on the topological magnetism and TLHE.” Ref. [70] was added. Table S1 and Fig. S9 were included in the Supplementary Information.